# Boosting Neurogenesis as a Strategy in Treating Alzheimer’s Disease

**DOI:** 10.3390/ijms26188926

**Published:** 2025-09-13

**Authors:** Abena Dwamena, Rashini Beragama-Arachchi, Hongmin Wang

**Affiliations:** Department of Pharmacology and Neuroscience, Garrison Institute on Aging, Center of Excellence for Translational Neuroscience and Therapeutics, School of Medicine, Texas Tech University Health Sciences Center, Lubbock, TX 79430, USA

**Keywords:** Alzheimer’s disease, adult hippocampal neurogenesis, neural stem cells, cognitive decline, regenerative therapy, epigenetic regulation, neurogenesis enhancement, neurotrophic factors, stem-cell therapy, dementia treatment

## Abstract

Alzheimer’s disease (AD) causes progressive cognitive decline and neuronal loss, partly due to the buildup of amyloid-β (Aβ) plaques and tau tangles. Despite years of research, treatments targeting these hallmark pathologies have yielded only modest clinical success, prompting interest in regenerative approaches to restore the brain’s ability to repair itself. One such approach focuses on adult hippocampal neurogenesis, the process by which neural stem cells (NSCs) produce new neurons throughout life. In AD, this process is impaired, worsening cognitive deficits. In this review, we examine the molecular pathways that control adult neurogenesis, including transcriptional, epigenetic, inflammatory, and metabolic mechanisms, and how they become dysregulated in AD. We also highlight various therapeutic strategies aimed at boosting neurogenesis, such as pharmacological treatments, stem cell therapy, gene therapy, and epigenetic modulation. Preclinical studies indicate that enhancing neurogenesis can improve cognition and reduce brain pathology in AD models. Several of these treatments are now being tested in clinical trials. Ultimately, promoting neurogenesis may offer a promising avenue to complement current AD therapies and help restore lost neural function.

## 1. Introduction

Alzheimer’s disease (AD) is a progressive, multifactorial neurodegenerative disorder characterized pathologically by the presence of two hallmark proteins, amyloid beta (Aβ) and hyperphosphorylated tau. Advances in research have evolved our understanding of Aβ toxicity, revealing that soluble Aβ oligomers, rather than insoluble plaques, are potent early drivers of synaptotoxicity and neuronal death in AD. These oligomers interfere with synaptic signaling and impair neuroplasticity independent of plaque formation [1,2,3,4]. As the disease progresses, soluble Aβ builds up in the extracellular space between neurons as plaques, while tau pathology appears inside neurons as neurofibrillary tangles that block axonal transport and communication. Over time, these pathological events cause damage to brain cells, affecting functions such as cognition, memory, decision-making, behavior, and personality [2,5]. AD affects over 55 million people worldwide, with projections indicating the number will triple by 2050 [6].

In recent years, adult neurogenesis has become an important and promising field in understanding and potentially mitigating neurodegeneration in AD. Adult neurogenesis is the process of generating new neurons from the neural stem cells (NSCs) or neural progenitor cells (NPCs) in the adult brain. In humans, it is restricted to specific regions of the brain: the subventricular zone (SVZ) and the subgranular zone (SGZ) of the hippocampal dentate gyrus [7,8]. The hippocampal dentate gyrus, a region integral to learning and memory, is among the earliest brain regions affected in AD [9].

Although growing evidence indicates that adult hippocampal neurogenesis (AHN) persists in humans, especially studies showing immature neuron markers in postmortem tissue from healthy individuals and those with AD, it remains a subject of debate. Initial research did not find strong evidence of neurogenesis in adult human hippocampi, raising questions about its prevalence and functional relevance [10]. However, methodological differences, including tissue preservation, detection sensitivity, and marker specificity, have likely contributed to these discrepancies.

Recent studies, utilizing advanced immunohistochemistry and transcriptomic methods, offer compelling evidence that AHN continues into old age and declines notably in AD [11,12]. This decline in neurogenic capacity may contribute to hippocampal dysfunction and memory deficits, raising the possibility that strategies aimed at restoring or boosting neurogenesis could help preserve cognitive function in AD. While the precise role of neurogenesis in AD pathogenesis remains under investigation, several preclinical studies suggest that boosting neurogenesis may ameliorate cognitive decline and neuronal loss, especially when combined with interventions targeting Aβ and tau pathology [9,13].

Despite efforts to eliminate amyloid-beta (Aβ) and tau in AD, clinical outcomes remain disappointing. For instance, drugs like Semagacestat and Bapineuzumab successfully reduced Aβ and tau levels, but failed to improve cognition, and in some cases, worsened it [14,15]. More recently, Semorinemab, an anti-tau antibody, was discontinued after showing no meaningful clinical benefit. These failures highlight a critical insight that clearing pathological proteins may not be enough to restore brain function [16,17]. As such, researchers are now exploring regenerative strategies that go beyond protein removal. Emerging approaches aim to reactivate neural stem cells, restore neurogenesis, and repair hippocampal networks using pharmacological, genetic, and epigenetic tools. By targeting the brain’s innate capacity for repair, these strategies offer a new path to affect AD treatment.

This review aims to provide a comprehensive overview of adult neurogenesis in the context of AD and examine therapeutic strategies, including pharmacological agents, molecular targets, and lifestyle interventions that aim to boost neurogenesis to mitigate AD pathology and cognitive impairment.

## 2. Overview of Adult Neurogenesis

Adult neurogenesis is the production of new neurons from neural stem cells (NSCs) or neural progenitor cells (NPCs) in the adult brain. Contrary to the long-held belief that neurogenesis occurs only during prenatal and early postnatal development, robust evidence now indicates that the mature brain retains neurogenic potential throughout life. In humans, it is restricted to the subventricular zone (SVZ), lining the lateral ventricles and the SGZ of the dentate gyrus within the hippocampus [18,19,20]. In the SVZ, neural stem cells give rise to neuroblasts that migrate to the olfactory bulb via the rostral migratory stream [21], whilst in the SGZ, progenitor cells generate granule neurons that integrate into existing hippocampal circuits [20]. The neurogenic process involves multiple stages: activation of quiescent NSCs, which can be triggered by extrinsic signals such as fibroblast growth factor-2 (FGF-2) or brain-derived neurotrophic factor (BDNF), followed by proliferation, lineage commitment, differentiation into neurons, migration, maturation, and finally, integration into neural circuits [22].

Neurogenesis is regulated by complex signaling networks involving growth factors, transcription factors, and epigenetic mechanisms. Key regulatory factors include brain-derived neurotrophic factor (BDNF), insulin-like growth factor-1 (IGF-1), vascular endothelial growth factor (VEGF), and Wnt signaling pathways [20]. The transcription factors Sox2, Tbr2, and NeuroD1 coordinate the progression from stem cells to mature neurons, while epigenetic modifications by DNA methyltransferases and histone-modifying enzymes fine-tune gene expression programs [23] (Figure 1).

In addition, various endogenous and environmental factors also significantly influence neurogenesis rates. Physical exercise, environmental enrichment, caloric restriction, and learning experiences promote neurogenesis, whereas chronic stress, aging, and inflammation inhibit it [24] (Figure 2 and Figure 3). These regulators provide an environment that supplies blood vessels, nutrients, and signaling factors to support cells involved in neurogenesis. The hippocampal dentate gyrus is now widely recognized as the primary site for persistent neurogenesis in the adult human brain.

## 3. Regulation of Adult Neurogenesis

The formation of new neurons is a tightly regulated process controlled by a complex interaction of internal molecular pathways and cellular dynamics, as well as the surrounding external environment. It begins with the activation of quiescent NSCs located within the SVZ or SGZ. Once activated, they proliferate to produce neuroblasts, which then differentiate and express markers such as doublecortin (DCX). As they mature, they migrate to the granule layer or olfactory bulb and eventually incorporate into existing neural circuits, where they participate in learning and memory. This process is governed by specific factors, which can either be intrinsic to the cells or affected by the environment in which the cells interact. The intrinsic factors are the main drivers influencing neurogenesis, as they determine the ability of neural stem cells to self-renew, differentiate, or stay quiescent. If this intrinsic system fails, no support from the extrinsic environment can effectively make up for it in neurogenesis [25].

In neurodegenerative diseases like AD, subtle dysregulation of these first-line regulators, ranging from transcriptional repression toward inappropriate protein clearance or mitochondrial dysfunction, can tip the balance toward NSC or NPC quiescence, senescence, or apoptosis. It is thus of utmost importance to identify these factors and develop targeted therapeutic modalities that may rescue or enhance the brain’s capacity for neurogenesis in the aging or degenerating brain. Below, we summarize some intrinsic factors that affect neurogenesis and consider how AD may affect them (Figure 1).

### 3.1. Transcription Factors and Intrinsic Regulators

Transcription plays a crucial role in the fate and function of neural stem cells throughout neurogenesis. This is made possible through an intricate network of transcription factors that act to either activate or suppress genes involved in the process of neurogenesis. Many intrinsic transcriptional factors, including Sox2, KIf9, NFIA, the STAT3-MYC axis (under specific developmental conditions), Yap1, and TLX, have been discovered to promote neurogenesis [26,27,28,29,30] (Figure 1).

Sox2 maintains the self-renewal and undifferentiated state of NSCs, sustaining the NSC pool for continuous neurogenesis [31]. In adult hippocampal neurogenesis, Sox2 expression in radial glia-like NSCs is essential for their survival. Ablating Sox2 results in the loss of NSCs and a reduction in new neuron production, as demonstrated by conditional knockout experiments [28].

The transcription factor Klf9 has emerged as an important regulator of neurogenesis that maintains stem cell quiescence. Guo et al. (2022) found that quiescent NSCs express high levels of KIf9, and when deleted conditionally in the adult dentate gyrus, it triggers stem cell activation and increased proliferation, transiently boosting neurogenesis but risking long-term stem cell exhaustion, highlighting the need for tight transcriptional control [32].

NFIA (Nuclear Factor I A) promotes neural progenitor lineage development and NSC maintenance. Recent studies reveal that knocking out NFIA in adult NSCs significantly decreases new neuron production and impairs hippocampus-dependent memory [30]. Without NFIA, memory deficits occur along with lower levels of NSCs, neuroblasts, and fully differentiated neurons, indicating a primary defect in NSC maintenance and highlighting the essential role of NFIA in AHN and cognition.

The STAT3 and MYC pathway involves STAT3 and c-MYC in controlling the proliferation and differentiation of neural stem cells (NSCs). Excessive STAT3 activity can cause NSCs to shift into a quiescent or gliogenic state, which hampers neurogenesis. A 2025 study found that high STAT3 activity sequesters MYC, stopping its neurogenic function and leading to abnormal neurogenesis. Since MYC is essential for progenitor cell proliferation and neuron formation, elevated STAT3 levels suppress MYC’s neurogenic role. Inhibiting STAT3 signaling, such as with artesunate in FMRP-deficient mice, has been shown to restore normal neurogenesis. Thus, a balanced STAT3–MYC axis is vital; MYC encourages neurogenic proliferation, while high STAT3 levels inhibit it [29].

Yap1, also known as Yes-associated protein 1, helps activate adult neural stem cells (NSCs). Its activity is higher in active NSCs, and deleting Yap1 reduces their numbers. Yap1 is crucial for transitioning NSCs from quiescence to proliferation. Overexpression can stimulate quiescent NSCs to re-enter the cell cycle, but excessive activation may cause abnormal differentiation, emphasizing the importance of upstream regulation. Overall, Yap1 promotes neurogenesis, and its reduction impairs this process, potentially leading to glioblastoma-like gene expression patterns in NSCs [27]

TLX (NR2E1), an orphan nuclear receptor, maintains neural stem cells (NSCs) in an undifferentiated and self-renewing state. It inhibits cell-cycle inhibitors like p21 and PTEN while activating genes that encourage proliferation, ensuring a continuous supply of progenitors for ongoing neurogenesis [26,33].

For these transcription factors, AD can disrupt their normal function. For instance, chronic stress and inflammation in AD can elevate Kif9 (a glucocorticoid-inducible gene) to continually suppress neurogenesis. Likewise, deficits in growth factor signaling (e.g., IGF-1 resistance in AD) could impair pathways like STAT3-MYC or Sox2 maintenance [31]. In summary, multiple transcription factors tightly coordinate the neurogenic process, and their dysregulation (through genetic or extrinsic means) likely underlies the reduced neurogenesis observed in Alzheimer’s disease.

Several external factors can influence the expression of intrinsic factors, either supporting or hindering the process of neurogenesis. For example, physical activity increases hippocampal levels of Wnt and BDNF, which can activate transcription factors such as Sox2 and TLX [34,35,36]. Similarly, environmental enrichment and learning stimulate neurogenic signaling pathways that often target common components, such as CREB [37]. Antidepressants, including selective serotonin reuptake inhibitors (SSRIs), promote neurogenesis through multiple mechanisms, including glucocorticoid receptor activation [38] and serotonergic pathways that involve CREB and BDNF [39,40]. Together, these examples demonstrate how lifestyle choices and pharmacological treatments can modulate key upstream factors involved in neurogenic transcription, offering potential avenues for therapy in AD.

In AD, various stressors, such as chronic neuroinflammation, oxidative stress, and dysregulated epigenetics, can disrupt transcriptional regulation. This includes the abnormal activation of factors like Klf9, which may remain persistently upregulated and ultimately inhibit neurogenesis (Figure 1). This highlights the importance of transcriptional dysregulation in the progressive loss of neurogenesis observed in AD [32].

### 3.2. Epigenetic Regulation

Epigenetic drift is an age-related and random change in gene expression that occurs without altering the underlying DNA sequence, impeding neurogenesis. These modifications, including DNA methylation and histone modification, regulate whether a gene is turned on or off (Figure 1). In aging and AD, disturbed epigenetic drift can decrease the expression of genes involved in neurogenesis. Blanco-Luquin et al. demonstrated this using a human in vitro model of adult hippocampal neurogenesis, showing that Aβ increases DNA methylation at the NXN gene, especially during early stages of neuron formation. The NXN gene is a redox regulator that protects NSCs and neurons from oxidative stress and regulates differentiation. This hypermethylation during early neuron formation led to decreased NXN expression, suggesting that AD-related Aβ may impair brain repair by silencing essential genes needed for new neuron formation [41]. Zocher et al. further showed that deleting the de novo methyltransferases (DNMTs), Dnmt3a and Dnmt3b, specifically in adult hippocampal NSCs, disrupted neuronal maturation. Although early neurogenesis remained unaffected, the resulting neurons exhibited poor dendritic development, fewer synaptic connections, and failed to integrate into hippocampal circuits, resulting in cognitive deficits [42]. Overall, these studies reveal that both Aβ toxicity and inherent methylation deficits interfere with epigenetic programs crucial for neuronal maturation.

### 3.3. Mitochondrial Dysfunction

A major abnormality in AD is mitochondrial dysfunction. Mitochondria act as the cell’s energy producers, generating ATP crucial for energy-intensive processes like neurogenesis. Interestingly, research indicates that before NSCs differentiate, they mainly rely on glycolysis for energy supply. During differentiation, however, they shift to depend more on oxidative phosphorylation, a process controlled by mitochondria [43]. This metabolic shift to mitochondrial respiration is essential for supporting the growth and maturation of new neurons.

In AD, pathological proteins like tau and Aβ are reported to infiltrate the mitochondrial matrix and bind to key proteins of the electron transport chain, including complex I. Aβ can be imported into mitochondria via the translocase of the outer membrane (TOM) complex and subsequently accumulates within mitochondrial cristae and disrupting mitochondrial integrity [44,45]. These interactions disrupt electron transport and increase reactive oxygen species (ROS) production, leading to mitochondrial damage [45]. In addition to ROS, factors such as maladaptive changes in microRNAs, chronic metabolic stress, and epigenetic alterations further exacerbate mitochondrial injury [46]. This damage causes mitochondrial fragmentation and functional decline. Increasing evidence indicates that mitochondrial dysfunction impairs NSCs’ ability to sustain the neurogenic niche and produce new neurons [47] (Figure 1).

Han et al. conducted a study on an AD-related stress model by depriving adult rat hippocampal neural progenitor cells of insulin (mimicking impaired insulin signaling as seen in AD brains). Their temporal microRNA profiling revealed significant upregulation of these miRNAs: miR-150-3p, miR-323-5p, and miR-370-3p, which induced mitochondrial fragmentation and ultimately led to neural cell death. Through miRWalk analysis and functional studies, the authors identified OPA1 and MFN2, which are regulators of mitochondrial fusion, as direct targets, where their levels markedly decreased. These results highlight a novel mechanism by which miRNA-induced mitochondrial dysfunction impairs hippocampal neurogenesis in AD [48].

When mitochondrial dynamics lean towards excessive fragmentation due to stress and Aβ/tau toxicity, NSCs cannot maintain the energy production or metabolic flexibility needed for neurogenesis, leading to reduced proliferation and survival. However, relying solely on stimulating systemic insulin release through glucose intake as a therapy is probably unsafe and ineffective. Although insulin is crucial for neuronal metabolism and mitochondrial function, increasing peripheral insulin levels by consuming high amounts of glucose can lead to hyperinsulinemia and insulin resistance. These states are linked to oxidative stress, inflammation, and cognitive decline, which impair neurogenesis and may worsen AD pathology [49,50,51].

In contrast, targeted activation of brain insulin/IGF-1 signaling looks more promising. Intranasal insulin delivery, which bypasses peripheral circulation and delivers insulin directly to the brain, has been shown in multiple preclinical AD models (including intracerebrovascular streptozotocin rats) to improve cognitive function, reduce tau hyperphosphorylation, decrease neuroinflammation, and promote hippocampal neurogenesis without affecting peripheral glucose or insulin levels [52]. Clinical trials and meta-analyses also suggest that intranasal insulin can modestly improve cognition in patients with mild cognitive impairment or AD, with good safety and short-term tolerability [53]. Together, these findings suggest that enhancing insulin signaling directly in the central nervous system, rather than depending on systemic glucose-driven insulin, provides a safer and more targeted approach for maintaining mitochondrial integrity and supporting neurogenesis in AD.

### 3.4. Synaptic Activity and Neurotransmitters

Synaptic degeneration is a key feature of AD, caused by the accumulation of amyloid plaques and tau tangles. Aβ oligomers directly bind syntaxin 1a and disrupt SNARE-mediated vesicle release, leading to impaired neurotransmission [54]. Neurotransmitters such as acetylcholine (ACh) and glutamate are especially impacted, leading to cognitive decline. In the early stages of AD, cortical and hippocampal circuits often become hyperactive before shifting to hypoactivity. This biphasic pattern has been associated with glutamate imbalance, abnormal intracellular calcium levels, and excitotoxic stress [55].

Enhancing synaptogenesis is crucial for cognition, and impairing this process leads to memory deficits. In a study to determine whether newly generated neurons could improve memory in AD, the authors first increased adult hippocampal neurogenesis by conditionally deleting Bax, an apoptotic gene. They found that this deletion boosted the survival and integration of adult-born neurons into memory circuits after labelling and analyzing the engram cells. They also observed an increase in dendritic spines, indicating restored synaptic connectivity within the dentate gyrus. By silencing the neurons with chemogenetics, the opposite effect was seen, suggesting that enhanced neurogenesis improves synaptic integration into the hippocampal circuit and thus boosts memory [56].

Neurotransmitter imbalances in AD also affect the development and survival of adult-born neurons. Ach, for example, is significantly decreased in AD, and the cholinergic hypothesis remains a key explanation for early cognitive decline. Nitenson et al. studied how ACh influences neurogenesis using an olfactory learning model. Pharmacological blockade of Ach activity with scopolamine impaired memory and learning and reduced the survival of adult-born neurons in the olfactory bulb. Conversely, the activation of ACh activity with physostigmine improved performance and neurogenic integration. Since basal forebrain cholinergic neurons connect to both the hippocampus and olfactory bulb and are among the first affected in AD, this research highlights a mechanistic link between cholinergic atrophy, impaired neurogenesis, and sensory or cognitive deficits. These behavioral effects were closely linked to neurogenesis: reducing ACh decreased the number of new neurons that survived and integrated into olfactory circuits, while increasing ACh activity enhanced their survival and integration [57,58].

GABA (gamma-aminobutyric acid) acts as an excitatory agent in immature neurons, aiding their development. In AD, GABA levels drop. Meanwhile, glutamate levels become dysregulated, causing excitotoxicity and stressing stem cells. A study involving memantine, which blocks NMDA receptors, showed improved dendritic growth and boosted neurogenesis [59]. These observations suggest that synaptic activity and neurotransmitters positively influence neurogenesis (Figure 1).

## 4. Role of Neurogenesis in Cognition and Memory

The brain is the primary organ responsible for cognition. Cognition broadly encompasses perception, learning, memory, emotions, attention, and decision-making. These intricate processes arise from dynamic interactions among various brain regions; notably, the hippocampus is involved in learning and memory, the neocortex handles higher-level processing, the amygdala is associated with emotional memory, and the prefrontal cortex oversees working memory and executive functions.

Adult-born neurons play a role in shaping hippocampal circuits that support cognition. The hippocampus is essential for creating new episodic memories of specific events and experiences, originating from the dentate gyrus. Its circuit is a unidirectional and trisynaptic excitatory pathway. The entorhinal cortex supplies the main input to the dentate gyrus’s granule cells through the perforant pathway. These granule cells send mossy fibers to CA3 pyramidal neurons, which in turn connect to CA1 neurons via Schaffer collaterals. CA1 then relays outputs to the subiculum and returns signals to the entorhinal cortex. This network demonstrates substantial synaptic plasticity, allowing it to adapt and integrate with existing networks [60].

These new neurons enhance the brain’s ability for pattern separation, which is the capacity to distinguish similar inputs, a key function for learning and memory. Evidence supports this role. For example, Clelland et al. showed that reducing adult-born neurons through irradiation or temozolomide treatment impaired mice’s ability to differentiate between nearby spatial locations in both touchscreen and maze tests. Conversely, increased neurogenesis by genetically deleting Bax, a pro-apoptotic protein, in neural progenitors of the dentate gyrus caused the mice to perform better on contextual and spatial discrimination tests. Together, these studies suggest that newborn neurons contribute directly to how the brain learns and adapts, and that this process begins to decline early in AD [61,62].

Beyond memory, decreased neurogenesis has been linked to mood disorders, stress sensitivity, and limited cognitive flexibility. Adult neurogenesis supports brain adaptability by producing new neurons within stable networks, enabling the brain to learn new skills without losing old memories [56,63,64,65,66]. Since AD affects the hippocampus early on, there is growing evidence that boosting neurogenesis might serve as a new adjunct to current treatments. However, it remains an open question whether increasing neurogenesis can effectively address hippocampal dysfunction in AD, and this area has been underexplored. Gaining insight into how neurogenic capacity is regulated and its influence on cognitive functions in the diseased brain could open new therapeutic avenues.

## 5. Impairment of Neurogenesis in AD

AD is closely linked to aging. Both natural aging and AD involve a decline in adult hippocampal neurogenesis. Although it is currently known that adult neurogenesis continues to progress throughout life, its rate and effectiveness decline with age and in neurodegenerative diseases [20,67].

Evidence from a quantitative and age-related study in rodents showed a decline in neural progenitor cell proliferation in the dentate gyrus of the rat brain, although proliferation did not completely stop. This indicates that the adult brain still has some neuroregenerative ability even at an advanced age [67]. Human post-mortem studies showed that the brains of healthy individuals, even at age 90, had evidence of immature neurons in the dentate gyrus, which decreased with age from 40 to 90 years. In AD post-mortem brains, this number steadily declined as AD advanced, and the decline was faster than in healthy subjects, regardless of age [11]. In individuals in the early stages of AD (Braak stages I–II, when pathology first begins), the number of DCX^+^ immature neurons in the dentate gyrus was notably lower than in age-matched healthy controls. This decline persisted and worsened as AD pathology progressed [11].

Single-nucleus RNA sequencing of hippocampal tissue from healthy individuals from week 20 of gestation up to 92 years showed the presence of immature granule cells persisting at all ages, although their numbers steadily decreased with age. The hippocampal tissue from AD patients exhibited a reduced level of immature granule cells compared to age-matched controls. Furthermore, these cells displayed altered gene expression profiles, indicating ongoing neuronal development. Despite this decline, immunohistochemistry confirmed the presence of rare but detectable neural progenitor cells in AD brains, suggesting that neurogenesis, though impaired, may still occur at very low levels [68].

## 6. Physiological and Pathological Modulators of Neurogenesis

### 6.1. Neuroinflammation

Neuroinflammation is a key regulator of neurogenesis, particularly in the context of AD. It influences all aspects of the neurogenic niche. It helps activate microglia, which are protective in healthy individuals. However, in AD, where chronic neuroinflammation occurs, microglia become permanently activated, leading to a reactive state that shifts them from the protective M2 state to the pro-inflammatory M1 form, which promotes inflammation and neuronal damage. The ongoing release of pro-inflammatory cytokines such as IL-1β, TNF-α, and IL-6 marks this shift. These cytokines hinder NSC and NPC proliferation, decrease neurotrophic factors like BDNF, and trigger cell cycle arrest and apoptosis, while preferring glial differentiation over neuronal development.

Chronic neuroinflammation in AD causes astrocytes to become reactive. They generate reactive oxygen species (ROS), complement proteins, and may obstruct synapse formation. Their reactive condition lowers the levels of vital trophic factors like BDNF, VEGF, and FGF-2, collectively starving NSCs and immature neurons of essential survival and differentiation signals. This overall creates a harmful environment that hampers neurogenesis by disrupting neuronal differentiation, maturation, and synaptic integration [69,70].

Evidence from AD mouse models supports this link. For example, minocycline (an anti-inflammatory antibiotic) treatment suppressed microglial activation and was associated with the preservation of hippocampal neurogenesis [71,72]. Similarly, animals treated with NSAIDs (non-steroidal anti-inflammatory drugs) like Ibuprofen or selective cytokine inhibitors have experienced recovery of neurogenesis, in addition to the restoration of cognition [73,74]. These results collectively reinforce the idea that a constant pro-inflammatory brain environment harms neurogenesis and that reducing neuroinflammation could provide a therapeutic approach to restore neurogenic capacity in AD. These results suggest that a pro-inflammatory niche is, in fact, an inhibitory niche.

### 6.2. Oxidative Stress

In AD, chronic cellular stress results from multiple pathological processes, including Aβ toxicity, tau accumulation, neuroinflammation, and mitochondrial dysfunction. Amyloid toxicity, for example, activates microglia and astrocytes, triggering a sustained neuroinflammatory response that increases the production of reactive oxygen species [75,76]. In vivo, Aβ oligomers activate the Nod-like receptor family pyrin domain-containing 3 (NLRP3) inflammasome in microglia, leading to IL-1β release and oxidative stress, contributing to synaptic and neuronal injury [77]. When ROS generation exceeds the capacity of the endogenous antioxidant defenses, oxidative stress damages macromolecules, including lipids, proteins, and DNA, impairing neural stem cell proliferation, differentiation, and survival [78].

This oxidative burden affects neurogenesis through both cell-intrinsic and extrinsic (niche) mechanisms. At the cellular level, ROS can directly harm cells by causing DNA damage, inducing senescence, or leading to cell death in NSCs [79]. Moreover, Aβ-induced ROS activate intracellular stress-response pathways, such as the p38 mitogen-activated protein kinase (MAPK) pathway, which is overactivated in early AD and contributes to synaptic dysfunction, neuronal apoptosis, and neuroinflammation. This pathway and can tilt the fate of NSCs towards glial formation or apoptosis, leading to the gradual depletion of the neurogenic pool and structural plasticity [80,81,82,83].

To counteract this process, recent studies have focused on enhancing endogenous antioxidant pathways. One promising strategy involves the activation of nuclear factor erythroid-2-related factor 2, a transcription factor known for its anti-inflammatory and antioxidant ability, which is being investigated as a potential therapeutic approach. It upregulates protective genes for cells involved in mitochondrial homeostasis, inflammation, and antioxidation. Nrf2 activators such as sulforaphane and dimethyl fumarate are being investigated for their ability to restore redox balance and protect NSCs [84].

### 6.3. Dysregulated Signaling and Growth Factors

Adult neurogenesis relies on a variety of growth and signaling factors that support the survival, differentiation, maturation, migration, and synaptogenesis of NSCs. This process is regulated by neurotrophic factors such as Brain-Derived Neurotrophic Factor (BDNF), Nerve Growth Factor (NGF), and Fibroblast Growth Factor (FGF), among others. These factors not only maintain the neurogenic niche but also influence key signaling pathways like Wnt/β-catenin, MAPK/ERK, and PI3K/Akt (Figure 2). Additionally, pathways like Wnt and Notch control stem cell behavior and plasticity.

In Alzheimer’s, their levels may decrease due to the combined effects of oxidative stress, tau pathology, Aβ, and chronic neuroinflammation, impacting their function. The hostile environment not only affects neurotrophic factors but also disrupts signaling (Figure 2).

Many studies have highlighted the multifaceted role of neurotrophic factors in shaping the neurogenic response under pathological conditions. For instance, the levels of BDNF were shown to decline as the severity of AD increases [85]. A reduced level not only impacts the proliferation of NSCs but also synaptic integration and maturation, which are essential for memory and learning [85]. In one study, BDNF therapy via hippocampal gene delivery in AD mouse models (APP/PS1 and rTg4510) increased BDNF expression, improving gene expression and behavior. In APP/PS1 mice, BDNF enhanced neuronal signaling (upregulating Camk2a) and reduced neurodegeneration signatures. Both models showed better spatial memory and plasticity. In rTg4510 mice, BDNF boosted neurogenesis and plasticity genes (Sox11, DCX, Neurod1) and decreased stress-related genes. Shared upregulated genes related to neurogenesis and neuroprotection included Npy, Crh, and Tac1. These results suggest BDNF affects disease-specific and conserved pathways to restore hippocampal function, offering a promising strategy to repair neurogenic niches and stabilize cognitive circuits in Alzheimer’s [86].

Other factors, such as IGF-1 (insulin-like growth factor 1), VEGF (vascular endothelial growth factor), and FGF-2, are also known to promote neurogenesis, and many of these are found at lower concentrations in AD brains due to general neurodegeneration and the loss of cells that produce them.

Beyond neurogenesis, signaling cascades like Wnt/β-catenin, MAPK/ERK, PI3K/Akt, and Notch Pathways are vital, supporting neuronal survival and function in healthy brains but becoming dysregulated in AD, impairing their roles. These pathways often operate downstream or intersect with neurotrophic factors. For instance, TrkB, activated by BDNF, stimulates PI3K/Akt and MAPK/ERK, influencing Aβ’s pro-apoptotic effects and dendritic growth. IGF-1 supports neurogenesis via neurotrophic mechanisms by activating PI3K/Akt and MAPK/ERK through IGF-1R. Mir et al. (2017) identified a new IGF-1–mediated pathway involving RIT1–Akt–Sox2, crucial for neural stem cell self-renewal and neurogenesis by upregulating Sox2, vital for stem cell maintenance and neuronal differentiation [31].

In AD, reduced IGF-1 availability and receptor are linked to impaired Akt signaling and lower Sox2 expression, which reduces neurogenic capacity. Therefore, therapeutic strategies that restore IGF-1 signaling may not only improve neuronal survival but also revitalize endogenous neurogenic niches affected by neurodegeneration.

In addition, many other factors influence neurogenesis, including exercise, age, and hormones, such as stress hormones, which are known to suppress neurogenesis [87,88,89,90] (Figure 3). With this understanding of mechanisms and modulators, therapeutics can be designed more rationally to counteract AD’s anti-neurogenic environment and restore a degree of regenerative capacity to the brain.

## 7. Therapeutic Strategies to Enhance Neurogenesis

Multiple strategies are being investigated to boost neurogenesis in AD. Given the evidence that neurogenesis is impaired in AD and may contribute to cognitive deficits, it is imperative to consider whether boosting neurogenesis can reverse the cognitive deficits seen in AD. There is growing interest in developing strategies to stimulate neurogenesis as a potential therapy.

Current research into regenerative therapies aimed at enhancing neurogenesis falls into several main categories: pharmacotherapy, stem cell-based therapies, genetic and epigenetic modulation, exosome-based delivery systems, and lifestyle interventions. These approaches aim not only to boost the proliferation and survival of neural progenitor cells but also to promote proper maturation, synaptogenesis, and integration into functional hippocampal circuits. Each method has its own set of advantages and limitations, and many are still in preclinical or early clinical stages of research.

### 7.1. Pharmacotherapy

Numerous studies have focused on using small molecules to stimulate neurogenesis. Therefore, pharmacotherapy remains a key approach in enhancing neurogenesis. Several classes of drugs have shown promise in preclinical and limited clinical studies.

#### 7.1.1. Acetylcholinesterase Inhibitors

Acetylcholinesterase inhibitors (AchEIs) like donepezil and galantamine are widely used in AD management due to their ability to increase acetylcholine levels and enhance cholinergic signaling, thereby improving cognition and memory. While AChEIs were initially developed to address neurotransmitter deficits, emerging evidence indicates potential neurogenic effects, primarily through indirect mechanisms.

For instance, a preclinical study demonstrated that chronic galantamine treatment reversed neurogenesis deficits in adolescent intermittent ethanol (AIE)-exposed rats by restoring DCX^+^ immature neuron levels, reducing apoptosis, and downregulating neuroinflammatory markers such as HMGB1, COX-2, and CCL2 [91]. These effects may stem from galantamine’s anti-inflammatory actions and its ability to stimulate α7-nicotinic receptors, which are known to influence neuronal survival and plasticity.

Similarly, donepezil has been shown to enhance CREB (cAMP Response Element-Binding Protein) activation in hippocampal neurons, leading to an increased expression of brain-derived neurotrophic factor (BDNF), a regulator of neuronal survival, differentiation, and synaptic plasticity [92]. In rodent models of vascular dementia, donepezil administration increased BrdU-labeled surviving neurons and improved spatial memory performance, suggesting a role in facilitating neurogenesis [93].

Although AchEIs primarily enhance cholinergic neurotransmission, studies indicate that they may also indirectly support adult neurogenesis, reducing neuroinflammation and increasing neurotrophic signaling (e.g., via BDNF). This broadens their relevance beyond symptomatic relief, suggesting that enhanced cholinergic signaling may facilitate neurogenesis-supportive pathways in AD.

#### 7.1.2. Proteostasis Enhancers

Dysregulated proteostasis hampers neurogenesis in AD. Compounds that boost proteostasis or autophagy may promote neurogenesis. Memantine, an NMDAR antagonist, increases 20S proteasome activity, clearing aggregation-prone proteins like p21, p27, p-Tau, and beta amyloid precursors, restoring neurogenesis in AD mouse models [94]. Fluorogenic assays showed increased caspase-like, chymotrypsin-like, and trypsin-like activity, indicating higher proteasome activity. Building on this, Stazi et al. (2020) found that chronic memantine treatment in the AD mouse model restored hippocampal neurogenesis, rescued CA1 neuron loss, and significantly improved cognition [59].

While the authors did not directly assess proteasome function, Sahin et al.’s findings suggest memantine’s neurogenic and neuroprotective effects may be partly due to improved proteostasis and clearance of aggregation-prone proteins, enhancing the neurogenic niche [94]. Similarly, metformin activates AMP-activated protein kinase, promoting autophagy and hippocampal neurogenesis, linked to gut microbiota modulation and decreased neuroinflammation [95].

#### 7.1.3. Mitogen-Pathway Inhibitors

Mitogen pathway inhibitors commonly used in cancer treatment can be repurposed to enhance neurogenesis by targeting shared growth pathways. A good example is trametinib, an MEK1/2 inhibitor used in oncology for melanoma, which was identified to be a potent inducer of adult NSCs. In vitro, trametinib treatment of adult NSCs increased levels of p15^INK4b^ and the proneuronal transcription factor Neurog2, thereby halting cell proliferation and shifting the balance toward neuronal fate commitment. When administered to AD mice, there was a marked increase in the levels of Sox2^+^/GFAP^+^ NSCs, Ki67^+^ proliferating cells, and DCX^+^ neuroblasts in the hippocampal dentate gyrus, SVZ, and the cortex [96]. Thus, oncology drugs can be repurposed to be used to boost neurogenesis.

#### 7.1.4. Phosphodiesterase Inhibitors

Modulating intracellular signaling pathways is another avenue. Phosphodiesterase (PDE) inhibitors, especially PDE4 and PDE5 inhibitors, are considered potential regulators of neuronal plasticity and neuroinflammation in neurodegenerative diseases. They cause the breakdown of cyclic adenosine monophosphate (cAMP) and cyclic guanosine monophosphate (cGMP), which enhances PKA/PKG signaling. This activation leads to CREB stimulation and the subsequent increase in BDNF levels, both of which are crucial for neuronal survival, synaptic plasticity, and cognitive function [97,98]. Preclinical studies have demonstrated that rolipram, a PDE4 inhibitor, promotes hippocampal CREB phosphorylation, reverses memory deficits, and boosts BDNF expression in AD models [99]. Although rolipram is not very widely used because of its side effects (like nausea), PDE4 inhibitors (such as roflumilast) or PDE9 inhibitors (that increase cGMP) are being explored for their cognitive enhancement properties [100,101]. Likewise, PDE5 inhibitors (commonly used for erectile dysfunction), like sildenafil, have shown some cognitive benefits from preclinical models, potentially through enhancement of cerebral blood flow, decreasing oxidative stress, and activating pro-survival signaling pathways, indirectly fostering a neurogenesis-friendly environment [102,103].

Although PDE inhibitors do not directly promote neural progenitor proliferation, they help reduce damaging microenvironmental factors like oxidative stress and inflammation. Additionally, they enhance neurotrophic signaling, which supports adult hippocampal neurogenesis and improves cognitive resilience in AD.

#### 7.1.5. Antidepressants (SSRIs)

Selective serotonin reuptake inhibitors (SSRIs) have been found to increase adult neurogenesis significantly. For example, chronic administration of fluoxetine was shown to induce neurogenesis in the adult hippocampus, a process that depends on 5-HT3 receptor signaling. This study confirmed that this receptor is only present on radial glial-like neural stem cells (Nestin^+^/GFAP^+^) and immature neurons (DCX^+^) in the dentate gyrus. Pharmacological or genetic deletion of this receptor prevented the increase in stem cell proliferation and neuroblast maturation caused by fluoxetine [104].

A systematic review of the therapeutic potential of fluoxetine on cognitive decline in AD showed that fluoxetine increased hippocampal neurogenesis, reduced Aβ and tau pathology, and improved synaptic plasticity via BDNF-CREB and anti-inflammatory pathways (NF-κB inhibition), regulated oxidative stress (Nrf2), with improvement in cognition [105]. Therefore, SSRIs are candidates for combination therapy in AD (targeting both mood and plasticity).

#### 7.1.6. Glucagon-like Peptide-1 (GLP-1) Agonists

Initially developed for type-2 diabetes, GLP-1 agonists are now being explored as potential disease-modifying therapies for AD. They mimic glucagon-like peptide-1 (GLP-1), which regulates blood sugar by suppressing glucagon and boosting insulin. Besides metabolic effects, GLP-1 receptors are present in the hippocampus, cortex, and other brain regions, affecting neuroplasticity and cell survival.

In emerging preclinical AD studies, it has been shown that GLP-1 agonists, particularly Liraglutide and Semaglutide, increase adult neurogenesis, decrease Aβ buildup, and enhance synaptic function [106]. These drugs also reduce microglial activation and shift the neuroimmune environment toward an anti-inflammatory, pro-neurogenic phenotype. They support BDNF signaling and protect against neuronal apoptosis, thereby creating a favorable environment for neuronal survival and maturation.

Liraglutide has been shown to cross the blood–brain barrier and support brain glucose metabolism, structure, and network connectivity in small-scale clinical trials involving patients with mild cognitive impairment or early AD. Semaglutide is currently being evaluated in the large-scale EVOKE and EVOKE Plus Phase 3 trials (NCT04777409, NCT04777396) for early AD. These drugs are gaining attention as potential disease-modifying agents through metabolic and neurogenic pathways [107,108,109]. If these trials demonstrate significant cognitive benefits, this class of drugs could mark a major shift in AD therapy, transitioning from managing symptoms to modifying the disease by increasing neurogenesis.

### 7.2. Stem Cell Therapies

A direct way to replace lost neurons is to transplant exogenous stem cells or stimulate endogenous ones in the brain. Unlike symptom-only therapies, stem cell strategies aim to regenerate tissue, promote repair, and possibly reverse cognitive decline.

A study showed that embryonic stem cells (ESCs) transplanted as medial ganglionic eminence (MGE)-like progenitors can effectively integrate into hippocampal circuits and restore learning and memory in mice with induced cognitive impairments. These cells matured into functional cholinergic and GABAergic neurons, neurotransmitters most impacted in AD. The treated mice exhibited notable improvements in memory and learning tests, supporting the idea that stem cell-derived neurons can help recover cognitive functions [110].

Ji et al. examined the therapeutic potential of induced neural stem cells (iNSCs), derived from fibroblasts, in APP/PS1 mouse models. They found that transplanted iNSCs survived, migrated, and differentiated into glial cells and neurons in vivo. Importantly, these mice demonstrated improved cognitive function and decreased amyloid pathology. Overall, this study underscores the promising role of iNSCs in Alzheimer’s treatment [111].

Another promising approach involves mesenchymal stem cell (MSC) therapy. Brody et al. demonstrated the safety and biological activity of intravenous Lomecel-B in mild AD, with increased anti-inflammatory and angiogenic markers in a phase 1 trial. While no direct neurogenesis markers were assessed, transient hippocampal volume gains and VEGF upregulation suggest potential pro-neurogenic effects. These results highlight the possibility that MSCs may enhance endogenous neurogenesis, and future studies should explore whether MSC therapies stimulate endogenous neural stem cell activity in humans [112].

Although stem cells have great potential for promoting neurogenesis, several challenges must be addressed before they can be safely and widely used in AD treatment. Preventing immune rejection is crucial in cell therapies, especially with allogenic transplants. Using autologous or iPSC-derived cells may help reduce this risk. Tumorigenicity remains a concern with the uncontrolled division of pluripotent cells. Most current trials use lineage-committed cells with limited proliferation to mitigate this. Additionally, due to the brain’s complex circuitry, simply adding a neuron does not ensure proper synapse integration for memory. Future research should focus on cell survival and functional integration into memory regions [113,114].

### 7.3. Genetic and Epigenetic Manipulation

#### 7.3.1. Genetic Manipulation

New gene therapy and editing advances enable in vivo brain cell manipulation, offering opportunities to boost neurogenesis by directly altering gene expression. These range from delivering neurotrophic genes to editing neural stem cell genomes and epigenomes.

One promising target is Neurotrophic factor-α1 (NF-α1), also known as carboxypeptidase E (CPE). Initially identified as a prohormone-processing enzyme, CPE was found to have neuroprotective and neurotrophic properties, especially in stress response. Prior studies have shown that this enzyme promotes neuronal survival through the activation of ERK and PI3/Akt signaling and may be involved in neurogenesis and synaptic maintenance. Building on this, Xiao et al. conducted a targeted in vivo hippocampal gene delivery of the enzyme in APP/PSI AD mice. The result was a significant reduction in Aβ accumulation, reversal of memory deficits, and the preservation of hippocampal neurons, highlighting the enzyme as a novel gene-based therapy with disease-modifying potential in AD [115].

Gene therapy focusing on brain-derived neurotrophic factor (BDNF) is an emerging research area. Delivery of a viral vector carrying the BDNF gene by injection into the entorhinal cortex and hippocampus of aged rats and monkeys led to increased synaptic density and enhanced memory performance in the animals. Currently, a clinical trial is underway to test BDNF gene delivery [116,117]. Should BDNF gene therapy prove safe and effective, it would not only validate neurotrophic gene delivery in AD but also address the neurogenic niche’s dysfunction.

Modern genome editing tools like CRISPR/Cas9 now enable precise gene manipulation of neurogenesis regulators, such as NTRK2, which encodes the TrkB receptor essential for activity-dependent neurogenesis in the adult hippocampus. Roussel-Gervais et al. used CRISPR/Cas9 to knock out NTRK2 in human neural progenitors, resulting in reduced neurogenesis and increased glial differentiation. These findings underscore TrkB’s role in directing neural progenitor fate towards neurons, suggesting that without TrkB, stem cells favor gliogenic differentiation [118].

Aging regulates NSC quiescence and neurogenic decline, with genome-wide screens unveiling new rejuvenation targets. In 2024, Ruetz et al. conducted CRISPR-Cas9 knockout screens in cultured and living NSCs from young and old mice, finding over 300 genes whose deletion reactivated aged NSCs, especially Slc2a4 (GLUT4). Knocking out Slc2a4 rejuvenated NSC activation in vitro and in vivo, doubling neuroblast production in aged mice. This study suggests that increased glucose uptake with age may inhibit NSC activation, and targeting metabolic pathways could restore neurogenesis in aged and AD brains [119].

#### 7.3.2. Epigenetic Modulation

Both aging and AD involve negative epigenetic changes like altered histone acetylation, DNA methylation, and chromatin remodeling, causing gene silencing crucial for NSC function. Drugs reversing these modifications can reopen chromatin, restore gene expression, and reactivate silenced NSC genes.

Histone deacetylase inhibitors (HDACis) like vorinostat and sodium butyrate are promising. HDACs remove acetyl groups from histones, causing chromatin to condense and silence genes. Inhibiting HDACs increases acetylation, activating chromatin. HDACs enhance memory in AD models. In a study, the HDAC inhibitor valproic acid enhanced hippocampal neurogenesis and object recognition in APP/PS1/Nestin-GFP triple transgenic AD mice, likely by increasing histone acetylation at genes for neurogenesis and learning [120]. Further supporting this, HDAC2 knockdown in CA1 pyramidal cells of APP/PS1 mice increased dendritic branching, mature spines, restored long-term potentiation, and improved memory. These outcomes highlight the therapeutic potential of epigenetic manipulations for neurogenesis, plasticity, and cognitive restoration in AD [121].

Beyond HDACs, DNA methyltransferase (DNMT) inhibitors such as 5-azacytidine promote neurogenesis and learning in mice, likely by reactivating silenced neurogenic genes. Likewise, histone acetyltransferase (HAT) activators, which promote gene expression via enhanced acetylation, restore plasticity and cognition in AD and tauopathy models [122,123,124].

### 7.4. Exosome-Based Therapies

A new approach to therapy uses exosomes, nanovesicles that are 30 to 150 nanometers in size, secreted from cells carrying proteins, lipids, and nucleic acids. They can cross barriers like the blood-brain barrier (BBB) and facilitate cell communication.

Exosome-based therapies shift from direct stem cell replacement to using cell-secreted components for brain repair. Mesenchymal stem cell-derived exosomes mimic MSC benefits, offering a safer, easier alternative to cell transplantation. Studies show MSC-derived exosomes promote brain repair, with preclinical models of stroke and AD demonstrating enhanced neurogenesis, improved synaptic function, and cognitive recovery after IV administration, often with fewer risks. A study by Chen et al. found that exosomes from Wharton’s jelly MSCs reversed memory deficits in AD mice, improving performance, glucose metabolism, and reducing Aβ, likely via miR-29a suppressing HDAC4. This highlights exosomes as promising AD therapy carriers [125].

Similarly, Liu et al. showed that bone-marrow MSC-derived exosomes improved cognitive performance, reduced inflammation, and increased the expression of BDNF and synaptic proteins. The therapy also boosted levels of DCX, a marker of newborn neurons, in the hippocampus, indicating enhanced neurogenesis. These findings suggest that exosome therapy may revive the brain’s repair systems by activating endogenous neurogenic pathways while reducing AD-related pathology [126].

Encouragingly, this preclinical success is translating into early-stage clinical trials. In a recent Phase I/II trial, Xie et al. showed that intranasal delivery of adipose-derived MSC exosomes in patients with mild-to-moderate cognitive impairment was safe and well-tolerated and resulted in improvements in cognitive scores over 36 weeks. Although there were no observed changes in amyloid burden, hippocampal atrophy was modestly slowed, indicating a neuroprotective effect. These findings represent some of the first evidence that exosome-based therapies could be practical, non-invasive treatments for AD [127].

Exosome therapy in Alzheimer’s is still preclinical but progressing rapidly, potentially offering treatment for AD. Challenges include dosing, as exosomes are heterogeneous, making standard isolation and reproducibility difficult. Although well tolerated, immunogenicity and clearance require careful monitoring, especially with repeated doses [128,129]. These studies suggest that exosome therapies may activate repair pathways, modulate disease mechanisms like inflammation and epigenetic issues, and open new regenerative options for AD.

## 8. Translational and Clinical Implications

Translating adult neurogenesis from bench to bedside is one of the most compelling new directions in the search for disease-modifying treatments for AD. Despite progress in targeting pathological proteins such as Aβ and tau to reduce neuropathology, these approaches have yet to produce meaningful cognitive recovery, highlighting a critical gap between biomarker reduction and functional restoration. Hence, the need for treatments that not only reduce pathology but also promote repair and restore cognition is of the essence.

Since adult hippocampal neurogenesis (AHN), mediated by NSCs, is strongly linked to memory and is impaired in AD, there is a need to utilize its restoration to treat the cognitive deficits and ensure repair and functional recovery.

### 8.1. Complementary Therapy

Several anti-amyloid therapies, including lecanemab, have demonstrated a reduction in Aβ plaques but have failed to effectively restore memory and executive function, creating a critical gap between biomarker modulation and functional recovery [130]. This underscores the need to shift focus beyond pathological clearance and toward functional regeneration. A combination approach could be ideal to remove toxic pathology and stimulate the brain’s repair mechanism simultaneously.

Preclinical studies support this analogy. For instance, BNN27, a small-molecule nerve growth factor mimetic, has demonstrated multimodal neuroprotective and neurogenic effects in 5XFAD AD mice with long-term use. It reduces Aβ plaques and gliosis, restores synaptic protein expression, and promotes adult hippocampal neurogenesis (AHN) by increasing BrdU and NeuN-positive neurons, as well as DCX^+^ immature neurons, compared to controls. These effects lead to improved performance in cognitive tasks. This dual-action profile indicates that drugs targeting both pathological and regenerative processes could provide better outcomes than treatments focusing on a single aspect [115,131]. Future trials could include neurogenesis-enhancing agents alongside Aβ and tau therapies to target pathology and network dysfunction, bridging biomarker success and cognitive benefit.

### 8.2. Identifying Biomarkers

A major challenge is the non-invasive monitoring of neurogenesis in humans. Finding biomarkers is crucial for translating regenerative therapies into clinical practice for AD. Traditional methods like BrdU labeling are impractical in humans [132]. Therefore, non-invasive indicators are needed. Some promising new biomarkers include non-invasive neuron-derived extracellular vesicles found in blood, which contain specific microRNAs such as miR-132 and miR-212. Levels of these microRNAs are reduced in early AD patients, reflecting decreased hippocampal neuroplasticity [133,134]. Developing and validating these biomarkers is essential for selecting patients, ensuring therapeutic efficacy, and tracking long-term outcomes of regenerative interventions.

### 8.3. Early Intervention

Therapeutic timing is critical. Neurogenic niches stay intact during mild cognitive impairment (MCI) and early AD but decline as the disease advances. Studies show neurogenic deficits come before cognitive symptoms, and stopping neurogenesis speeds up decline; while boosting it early improves memory [13,56].

Postmortem human studies reveal that DCX^+^/PCNA^+^ neuroblasts persist in MCI but decline sharply in late-stage AD, correlating with memory impairment and reduced hippocampal circuit integrity [135]. Therefore, early interventions targeting neuroregenerative mechanisms before the collapse of neurogenic potential may offer maximal therapeutic promise.

### 8.4. Potential Risks and Safety Considerations

While emphasizing benefits, it is essential to also consider risks. Excessive stimulation of neurogenesis may lead to abnormal circuit development or seizures, since newborn neurons are particularly hyperexcitable, which, if dysregulated, may increase network excitability and seizure susceptibility [136,137]. In addition, hippocampal hyperplastic states in mice can produce mania-like and hyperexploratory behaviors, so pro-neurogenic treatments need careful adjustment and should be stopped if adverse effects are observed [138].

For cell therapies, the risk of immune reactions, ectopic migration, or tumor formation is a risk [139,140]. So rigorous monitoring (MRI scans for any tumor formation, cognitive assessments for any unusual changes) will be needed.

In summary, the translational path for neurogenesis-based AD therapy is becoming clearer. Key strategies include early intervention, combination therapies, biomarker development, and safety optimization. As we move beyond amyloid and tau, targeting brain repair and neurogenesis offers promise to restore cognition and functional independence. This regenerative approach, when properly timed and monitored, could transform AD treatment from symptomatic relief to true circuit-level restoration.

## 9. Conclusions

AD treatment remains a challenge, but emerging evidence suggests that enhancing the brain’s regenerative capacity could improve therapeutic results. In this review, we emphasized that clearing pathological proteins alone is not enough to restore cognition in AD. Instead, a paradigm shift is happening, moving beyond just stopping neurodegeneration toward actively rebuilding neural circuits through increased adult neurogenesis.

Research shows that the adult brain can continue to form neurons throughout life. However, this ability is compromised early on by aging and neurodegenerative diseases, which prevent the integration of new neurons. By focusing on the mechanisms that control neural stem cell function, survival, and integration, we can bridge the gap between reducing neuropathology and enhancing memory and executive function. While preclinical results are promising, it remains uncertain whether significantly increasing neurogenesis in human AD brains can be achieved or is sufficient to provide clinical benefit. However, the evidence reviewed here strongly supports further investigation of this approach.

In conclusion, enhancing adult neurogenesis presents a promising new avenue in AD therapy, emphasizing the promotion of functional recovery. While significant challenges persist, particularly in ensuring timely, effective, and safe neurogenic stimulation in patients, the potential benefits for cognitive restoration and quality of life are substantial.

## Figures and Tables

**Figure 1 ijms-26-08926-f001:**
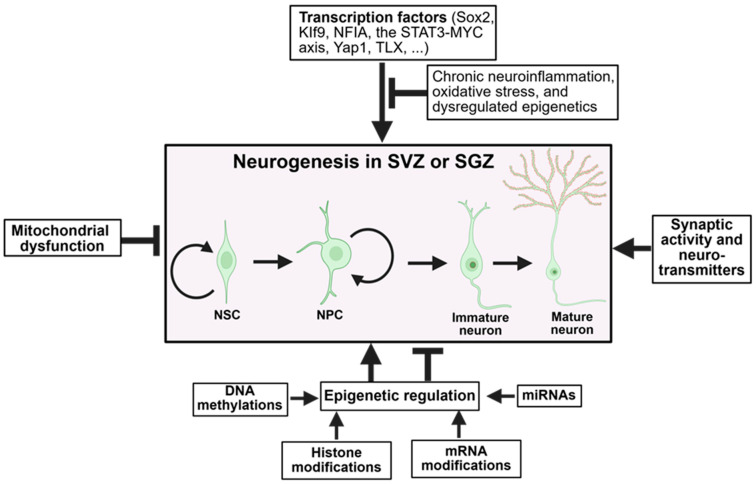
Cellular and molecular mechanisms that regulate adult neurogenesis. Many transcription factors, synaptic activity and neurotransmitters, and numerous epigenetic regulations enhance (indicated by arrows) neurogenesis in the adult brain, while mitochondrial dysfunction and some types of epigenetic regulations play the opposite role (indicated by the block arrows). NSC, neural stem cell; NPS, neural progenitor cell. The figure was made using BioRender (www.biorender.com).

**Figure 2 ijms-26-08926-f002:**
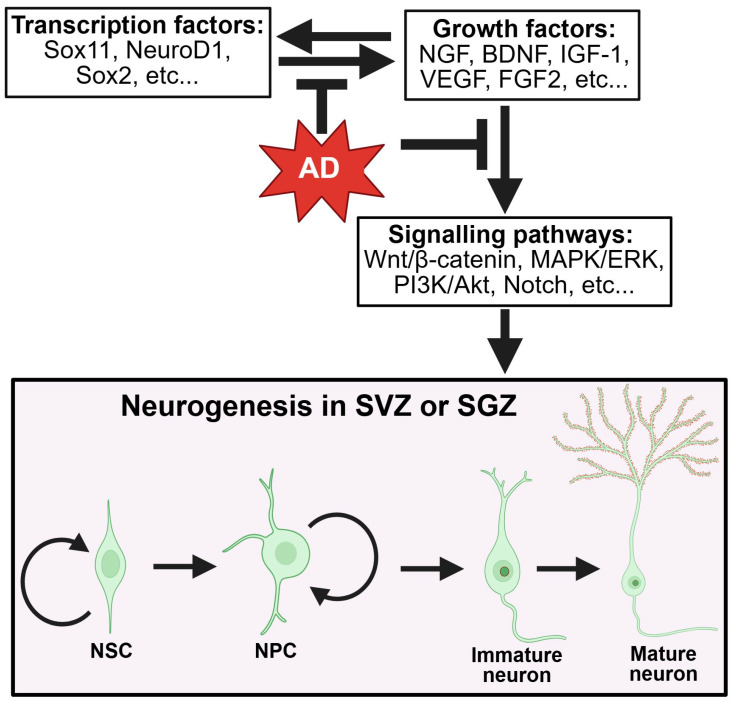
Growth factors in neurogenesis. Transcription factors, such as Sox11, NeuroD1, and Sox2, facilitate (indicated by arrows) the production of growth factors (and vice versa), including NGF, BDNF, VEGF, etc., which activate Wnt/β-catenin, MAPK/ERK, PI3K/Akt, and Notch signaling pathways to promote neurogenesis. AD inhibits (indicated by the block arrows) the transcription factors that regulate the growth factors, suppressing the related signaling pathways and leading to reduced neurogenesis. The figure was made using BioRender (www.biorender.com).

**Figure 3 ijms-26-08926-f003:**
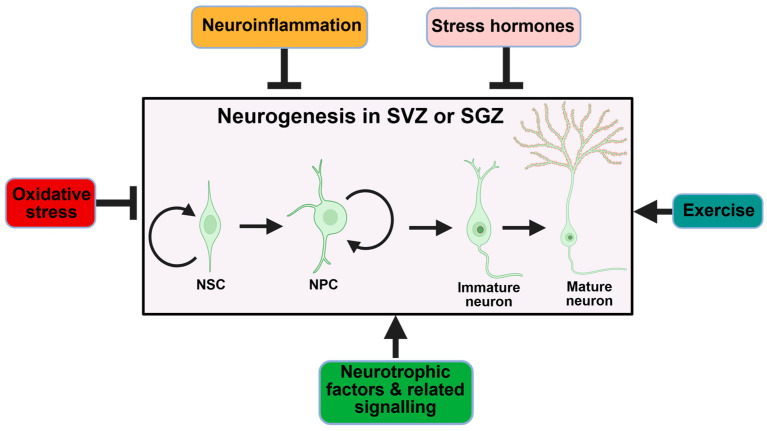
Physiological and pathological modulators of neurogenesis. In a physiological or pathological condition, neuroinflammation, oxidative stress, and stress hormones inhibit (indicated by block arrows) neurogenesishowever, exercise, neurotrophic factors, and their associated signaling pathways facilitate (indicated by arrows) neurogenesis. The figure was made using BioRender (www.biorender.com).

## Data Availability

Not applicable.

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
