# Peer review of "Boosting Neurogenesis as a Strategy in Treating Alzheimer’s Disease"

_ijms, 2025, doi:10.3390/ijms26188926_

Round 1
Reviewer 1 Report
Comments and Suggestions for Authors
The paper, titled "Boosting Neurogenesis as a Strategy in Treating Alzheimer's Disease," presented by Aben Dwamen, Rashini Beragama-Arachchi, and Hongmin Wang, provides a comprehensive, well-documented review of the literature on the role of adult neurogenesis in Alzheimer's disease (AD) and potential therapeutic strategies aimed at stimulating it. The authors discuss both the mechanisms regulating neurogenesis (transcriptional, epigenetic, mitochondrial, synaptic) and environmental and pathophysiological factors (inflammation, oxidative stress, signaling abnormalities). A detailed discussion of therapeutic strategies is provided, ranging from pharmacotherapy to cellular, gene, epigenetic, and exosome-based therapies. The text is clear, logically structured, and supported by up-to-date literature references (including recent studies). The authors also consider a broad spectrum of mechanisms and interventions, providing a comprehensive overview of the current state of knowledge.
In my opinion, this work is a valuable, up-to-date, and comprehensive literature review that may be of interest to both basic researchers and clinicians specializing in Alzheimer's disease. I recommend publication with minor revisions:
Introduction
Line 32 "Aβ aggregates form plaques externally in the spaces…..." This sentence is somewhat unclear. What do these plaques form aggregates externally?
- Overview of Adult Neurogenesis
Lines 77-83: The abbreviations NSCs, NPCs, SVZ, SGZ may be difficult for potential readers to decipher. Please explain them.
Lines 84-85 "The neurogenic process involves multiple stages: quiescent NSC activation…." Could the authors provide at least one example of the activating factor they mention?
In Part 3 (3.1. Transcription Factors and Intrinsic Regulators), the authors describe many factors that activate the neurogenesis process. They also cite external factors (such as stress) that can inhibit activating factors. However, there are no factors that are particularly interesting from the reader's perspective, which could influence the activation of factors that stimulate neurogenesis. Could you provide some examples?
- (3.3. Mitochondrial Dysfunction) The authors, citing the work of Han et al., suggest that a lack of insulin leads to mitochondrial degradation, which consequently negatively impacts the process of neurogenesis. Would, therefore, stimulating insulin secretion stimulated by glucose consumption be a good solution for maintaining mitochondrial integrity and activating neurogenesis?
Section 7.1.1. Acetylcholinesterase Inhibitors.
In their work, the authors focus on neurogenesis, or the process of forming new neurons. Frankly, AChE inhibitors such as donepezil, effectively used in Alzheimer's disease, cause (as the authors themselves admit) only upregulation of cholinergic signaling. However, they do not affect the formation of new neurons. Is the description of the use of acetylcholinesterase inhibitors justified in this work? Similarly, the phosphodiesterase inhibitors described in section 7.1.4.
Author Response
Response to Reviewers’ Comments
Reviewer 1
The paper, titled "Boosting Neurogenesis as a Strategy in Treating Alzheimer's Disease," presented by Aben Dwamen, Rashini Beragama-Arachchi, and Hongmin Wang, provides a comprehensive, well-documented review of the literature on the role of adult neurogenesis in Alzheimer's disease (AD) and potential therapeutic strategies aimed at stimulating it. The authors discuss both the mechanisms regulating neurogenesis (transcriptional, epigenetic, mitochondrial, synaptic) and environmental and pathophysiological factors (inflammation, oxidative stress, signaling abnormalities). A detailed discussion of therapeutic strategies is provided, ranging from pharmacotherapy to cellular, gene, epigenetic, and exosome-based therapies. The text is clear, logically structured, and supported by up-to-date literature references (including recent studies). The authors also consider a broad spectrum of mechanisms and interventions, providing a comprehensive overview of the current state of knowledge.
In my opinion, this work is a valuable, up-to-date, and comprehensive literature review that may be of interest to both basic researchers and clinicians specializing in Alzheimer's disease. I recommend publication with minor revisions:
Introduction
Line 32 "Aβ aggregates form plaques externally in the spaces…..." This sentence is somewhat unclear. What do these plaques form aggregates externally?
Response: Thank you for the comments. We apologize for the confusion. We have clarified that aggregates of Aβ accumulate in the extracellular space between neurons, forming amyloid plaques, while hyperphosphorylated tau aggregates within neurons, creating neurofibrillary tangles that interfere with nerve transport and communication. This makes it explicit that the plaques form outside neurons in the brain’s intercellular spaces, and removes any ambiguity.
2. Overview of Adult Neurogenesis
Lines 77-83: The abbreviations NSCs, NPCs, SVZ, SGZ may be difficult for potential readers to decipher. Please explain them.
Response: We agree and have defined all abbreviations upon first use in the main text. We have defined NSCs as neural stem cells, NPC’s as neural progenitor cells, SVZ as subventricular zone, and SGZ as subgranular zone.
Lines 84-85 "The neurogenic process involves multiple stages: quiescent NSC activation…." Could the authors provide at least one example of the activating factor they mention?
Response: We have added examples of extrinsic factors that lead to the activation of quiescent neural stem cells, such as fibroblast growth factor-2 (FGF-2) or brain -derived neurotrophic factor (BDNF). These are well-known pro-neurogenic factors that have been shown to initiate the neurogenic process.
In Part 3 (3.1. Transcription Factors and Intrinsic Regulators), the authors describe many factors that activate the neurogenesis process. They also cite external factors (such as stress) that can inhibit activating factors. However, there are no factors that are particularly interesting from the reader's perspective, which could influence the activation of factors that stimulate neurogenesis. Could you provide some examples?
Response: We agree with this suggestion. In view of this, we have expanded Section 3.1 to include specific examples of modifiable external factors that can inhibit transcription factors and intrinsic regulators. We made mention of factors such as physical activity, environmental enrichment, and certain drugs, which can modulate key upstream factors involved in neurogenic transcription, offering potential avenues for therapy in AD.
- (3.3. Mitochondrial Dysfunction) The authors, citing the work of Han et al., suggest that a lack of insulin leads to mitochondrial degradation, which consequently negatively impacts the process of neurogenesis. Would, therefore, stimulating insulin secretion stimulated by glucose consumption be a good solution for maintaining mitochondrial integrity and activating neurogenesis?
Response: This is an insightful question. While insulin/IGF-1 signaling benefits neuronal and stem cell health, simply increasing insulin through glucose uptake isn't an ideal therapy. It can cause hyperinsulinemia and insulin resistance, which damage the brain and may worsen oxidative stress and inflammation, further impairing disease outcomes. Instead, targeted strategies to enhance insulin/IGF-1 signaling in the brain are more promising. For example, intranasal insulin delivery has been shown to improve memory in patients. We thus acknowledge the reviewer’s suggestion and clarify that boosting insulin signaling, rather than just increasing glucose intake, is more beneficial.
Section 7.1.1. Acetylcholinesterase Inhibitors.
In their work, the authors focus on neurogenesis, or the process of forming new neurons. Frankly, AChE inhibitors such as donepezil, effectively used in Alzheimer's disease, cause (as the authors themselves admit) only upregulation of cholinergic signaling. However, they do not affect the formation of new neurons. Is the description of the use of acetylcholinesterase inhibitors justified in this work? Similarly, the phosphodiesterase inhibitors described in section 7.1.4.
Response: We understand the concern. Our reason for including these therapies is that, beyond their symptomatic effects, preclinical evidence indicates that these drugs can indirectly promote neurogenesis. We have clarified this in our revision by referencing studies where acetylcholinesterase inhibitors, such as Donepezil, activate Cyclic AMP Response Element-Binding protein (CREB), which increases Brain-derived neurotrophic factor (BDNF). This supports the survival and growth of new neurons, suggesting that these drugs may help create a neurogenic environment even if they do not directly induce neurogenesis.
Similarly, phosphodiesterase inhibitors (PDEs), such as rolipram, have been shown to enhance hippocampal neurogenesis and reverse memory deficits in an Alzheimer's disease (AD) model. We highlighted that modulating cAMP/cGMP signaling through phosphodiesterase inhibitors not only improves cognitive function but also promotes neurogenic processes in the brain. Moreover, newer PDE inhibitors are being investigated for their potential cognitive benefits with fewer side effects. Although primarily symptom-relieving, they can also support neurogenic capacity.
We hope this answers the concerns regarding the relevance of these sections.

Reviewer 2 Report
Comments and Suggestions for Authors
The manuscript examines the molecular mechanisms underlying neurogenesis and how they are disrupted in Alzheimer’s disease (AD). The authors provide evidence that enhancing adult neurogenesis could work as a promising therapeutic strategy for AD. Given the limited success of current treatments targeting amyloid-β plaques and tau tangles, the study explores how stimulating the brain’s innate ability to generate new neurons may help counteract cognitive decline. In conclusion, this work contributes significantly to the field by emphasizing neurogenesis as a complementary and innovative avenue for AD therapy. However, the authors should address several major concerns before the manuscript can be considered suitable for publication:
1 - On page 4, lines 133–144, the authors should provide more detailed information about the transcription factors. They only cite several factors involved without explaining their mechanisms of action. They should present them in the same way as they did for the factor Klf9.
2 - This topic would benefit from a figure/scheme showing the roles of the various growth factors. This figure could be included at the end of the manuscript, integrating the information on transcription factors with that of growth factors.
3 - Several paragraphs throughout the manuscript lack proper references to support the information presented, for example:
Page 5, lines 174-180
Page 5, lines 190-194
Page 7, lines 288-296
Page 7, lines 307 does not inform what NSAIDs is
Page 8, lines 314-330
Page 8, lines 339-345
Page 9, lines 363-372
Page 9, lines 377-380
Page 11, lines 443-449
Page 15, lines 656-659
Author Response
Response to Reviewers’ Comments
Reviewer 2
The manuscript examines the molecular mechanisms underlying neurogenesis and how they are disrupted in Alzheimer’s disease (AD). The authors provide evidence that enhancing adult neurogenesis could work as a promising therapeutic strategy for AD. Given the limited success of current treatments targeting amyloid-β plaques and tau tangles, the study explores how stimulating the brain’s innate ability to generate new neurons may help counteract cognitive decline. In conclusion, this work contributes significantly to the field by emphasizing neurogenesis as a complementary and innovative avenue for AD therapy. However, the authors should address several major concerns before the manuscript can be considered suitable for publication:
1 - On page 4, lines 133–144, the authors should provide more detailed information about the transcription factors. They only cite several factors involved without explaining their mechanisms of action. They should present them in the same way as they did for the factor Klf9.
Response: Thank you for your feedback. We have broadened our discussion on the main transcription factors involved in adult neurogenesis in Section 3.1. These updates help clarify how each transcription factor influences adult neurogenesis, similar to our explanation for Kif9.
2 - This topic would benefit from a figure/scheme showing the roles of the various growth factors. This figure could be included at the end of the manuscript, integrating the information on transcription factors with that of growth factors.
Response: A new figure (current Figure 2) is added to summarize the role of growth factors and the related transcription factors in neurogenesis. All figures are now located at the end of the manuscript.
3 - Several paragraphs throughout the manuscript lack proper references to support the information presented, for example:
Page 5, lines 174-180
Page 5, lines 190-194
Page 7, lines 288-296
Page 7, lines 307 does not inform what NSAIDs is
Page 8, lines 314-330
Page 8, lines 339-345
Page 9, lines 363-372
Page 9, lines 377-380
Page 11, lines 443-449
Page 15, lines 656-659
Response: We included citations to support the information in those sections and clarified the NSAID abbreviation. We believe these improvements address all of the concerns about missing citations and unclear terminology. Thank you for highlighting these issues, which helped us improve the rigor and clarity of our review article.

Round 2
Reviewer 2 Report
Comments and Suggestions for Authors
The authors have addressed all the suggestions provided, and the revised version of the manuscript has been improved with the incorporation of additional references and stronger support for the information presented. I therefore recommend acceptance of the article.
Author Response
In the first paragraph of the Introduction section, the importance of the two specific proteins of Alzheimer's disease, amyloid beta and hyperphosphorylated Tau, is recalled. The authors discuss the toxicity of Abeta aggregates and resulting amyloid plaques to brain cells. However, this partial and somewhat old vision completely overlooks the early responsibility of soluble Abeta oligomers for neuronal death, as well as the consequences on synaptotoxicity and loss of plasticity. This omission must be corrected and correctly referenced in the final manuscript.
Response:
We agree that soluble Aβ oligomers are critical early drivers of synaptic function and impaired plasticity. We have revised the introduction to include this suggestion.